# Reminders of Mortality: Investigating the Effects of Different Mortality Saliences on Somatosensory Neural Activity [note 1]

**DOI:** 10.3390/brainsci13071077

**Published:** 2023-07-16

**Authors:** Istvan Laszlo Gyimes, Elia Valentini

**Affiliations:** Centre for Brain Science, Department of Psychology, University of Essex, Colchester CO4 3SQ, UK; ig16036@essex.ac.uk

**Keywords:** mortality salience, existential anxiety, EEG, pain, somatosensory perception, event-related theta activity

## Abstract

The Terror Management Theory (TMT) offered a great deal of generative hypotheses that have been tested in a plethora of studies. However, there is a still substantive lack of clarity about the interpretation of TMT-driven effects and their underlying neurological mechanisms. Here, we aimed to expand upon previous research by introducing two novel methodological manipulations aimed to enhance the effects of mortality salience (MS). We presented participants with the idea of the participants’ romantic partner’s death as well as increased the perceived threat of somatosensory stimuli. Linear mixed modelling disclosed the greater effects of MS directed at one’s romantic partner on pain perception (as opposed to the participant’s own mortality). The theta event-related oscillatory activity measured at the vertex of the scalp was significantly lower compared to the control condition. We suggest that MS aimed at one’s romantic partner can result in increased effects on perceptual experience; however, the underlying neural activities are not reflected by a classical measure of cortical arousal.

## 1. Introduction

Ernest Becker argued that fear of death is an unlimited well of anxiety. Humans have buffered such existential anxiety by developing symbolic systems of meaning and value, such as cultures, religions, and belief systems. By subscribing to these, humans grant themselves an escape from mortality and thus find relief from the emotionally taxing existential anxiety [1]. Following up on Becker’s ideas, the Terror Management Theory (TMT) posits that humans are keener to defend their cultural views and increase their self-esteem when faced with the idea of mortality [2]. In two companion landmark papers [3,4], researchers developed an empirical approach to test the hypothesis that reminders of death would lead participants to praise or punish individuals that uphold or violate cultural values, respectively. The core experimental manipulation entailed simply submitting the participants to the Mortality Attitudes Personality Survey. This survey consisted of a brief ad hoc two-item, open-ended questionnaire whereby they were asked to write about (a) what will happen to them as they physically die and (b) the emotions that the thought of their own death arouses in them. As this methodology was intended to make mortality salient, it has been referred to as mortality salience (MS) manipulation. Importantly, TMT posits that MS effects take place only if participants are distracted from conscious reminders of death [5].

Since then, MS has been used in over hundreds of studies and the effects of mortality/death reminders, as operationalised according to the TMT, have been replicated over five-thousand times [6,7]. For example, the hypothesis that increased self-esteem helps an individual buffer the anxiety triggered by reminders of death is supported by several studies [2,8,9,10]. However, the TMT has seen rising criticism in the last years [11,12], and more recent attempts to replicate core findings failed [13,14].

Despite criticism, there has been a growing effort to investigate the neurological underpinnings of death cognition. A few studies implemented death-related cues without the reflective/contemplative procedure [15,16,17,18,19]. Notwithstanding the methodological variability, most of the findings supported the notion that reminders of death have specific effects on participants’ behaviour and neural activity. Yet, there is still confusion on their interpretation [20,21,22]. Our previous studies have shown an effect of MS on vertex neural responses to noxious stimuli [22,23]. The findings suggest a top-down modulation of MS on delta [22] and theta [23] amplitudes evoked by thermal nociceptive stimuli.

Noxious stimuli have a homeostatic/motivational value for the organism [24]. They signal a potential threat to the body and, as such, offer themselves as an elective tool to quantify the anxiogenic effects stemming from the thoughts of death. Painful stimuli are perceived as more threatening compared to non-painful stimuli. For example, they elicit a greater threat-related response compared to other sensory non-painful stimuli, as long as there is no competing pain-unrelated motivation [25], and they are harder to ignore [26]. Neuroscientific evidence supports the notion that both MS and somatic pain perception may share similar neural resources and thus substantiate the use of pain and somatosensory brain responses as dependent variables. Indeed, reminders of death convey very salient (though symbolic) information as they shove the idea of inevitable death into attentional focus. Functional Magnetic Resonance Imaging studies reported increased activity in the salience network (anterior cingulate, anterior insula, amygdala, and ventrolateral prefrontal cortex) after reminders of death [15,17,27,28]. This network is equally activated during processing of nociceptive stimuli and is highly modulated by top-down cognitive factors [29]. Further support for the notion of a physiological interaction between MS and pain constructs stems from the biological substrates of anxiety. Past research has suggested that the human body serves as a perpetual reminder of our finiteness [30]. Thus, the body is perfectly placed as an interface between symbolic threat (i.e., existential uncertainty) and sensory threat (i.e., body damage). It follows that the fear and anxiety concerning bodily harm or death may be rooted in a common anxiety biological system [20,31].

Because of these theoretical and empirical premises, our past work relied on the hypothesis that mortality salience interferes with cortical responses to painful somatosensory stimuli through the top-down allocation of attentional resources, in turn leading to heightened stimulus detection and attentional orientation processes [32].

The current study aimed to enhance MS effects by introducing two new methodological features. First, we added a new MS scenario where we asked participants to think about the death of their romantic partner instead of their own. The rationale for this manipulation is based on the tight link between existential anxiety and the need for intimacy [33]. According to TMT, when reminded of mortality people tend to strive for self-esteem [34]. Self-esteem can be achieved via several means, as an assimilation to cultural ideas (e.g., culturally preferred look, behaviour, principles, etc.). One of these cultural ideas is the concept of close relationships [33]. The idea of striving to foster close relationships when faced with mortality confirms the evolutionary tendency of looking for protection in others [35]. Further supporting the idea that affiliating to a group works as a defence against existential anxiety, Koole et al. [36] showed that even a brief touch on the shoulder can lessen existential anxiety. Close relationships may then act as ‘safe spaces’, soothing the individual anxiety originating from multiple existential sources including the idea of one’s own inevitable death [33,35]. If close relationships have such a protective function, the threat of a dear one’s death may as well elicit similar existential anxiety as the one elicited by the thoughts of one’s own death. Even more, according to Mikulincer et al., the looming idea of a romantic partner’s death may have a stronger impact than the idea of one’s own death because the “[…] separation from a relationship partner leaves people unprotected from the awareness of their mortality” (page 296) [37]. In keeping with this interpretation, we hypothesised that the idea of the death of one’s romantic partner would result in larger behavioural and neural effects than those triggered by the idea of one’s own death. Second, we introduced an additional psychological threat manipulation for noxious stimuli [38]. As mentioned above, expectation of somatosensory stimuli can increase the perceived pain, and more threatening stimuli are perceived as more painful. We assumed that more threatening stimuli would have been more effective in highlighting the increased physiological arousal triggered by MS effects compared with less threatening stimuli.

Here, we expand on our previous research involving somatosensory painful stimuli and MS manipulation [22,23] while exploring methodological means to increase effect size for studies involving physiological measures. Indeed, our previous work, while confirming an effect of MS at both perceptual and neural levels, revealed a significant modulation of theta and delta, and very late nociceptive evoked potentials triggered in the context of a fast stimulus repetition paradigm [22,23]. Here, we introduce a novel MS aimed at one’s romantic partner. This was expected to enhance the effects of MS, as it jeopardises a buffer against existential anxiety, namely, the romantic relationship [33]. Furthermore, we introduced more threatening stimuli because expectation of pain increases both the level of perceived pain and the pain-related neural responses [39], thus heightening the MS effects on perception and neural responses.

In the current study, we focus again on pain perception and event-related EEG theta oscillatory activity as proof-of-concept dependent variables to assess the changes associated with each independent variable. We chose to investigate vertex theta activity as it is one of the most classical sensory responses across several sensory modalities and displays optimal signal-to-noise ratio, apt to identify subtle cognitive and affective modulations of the cortical activity. As per our previous work, we focused on measuring MS-related effects on the theta component measured at the Cz electrode [23] but we also used a vertex region of interest (ROI) as well [40], thus controlling for type I error. We expected an increase in pain perception and vertex theta response after MS induction (compared with a negative control mindset). We expected this increase to be greater following one’s romantic partner MS. Additionally, we expected the MS effects to be greater for more threatening stimuli.

## 2. Materials and Methods

### 2.1. Participants

Twenty-five healthy participants who were not in an exclusive relationship (11 females, mean age 22.16 ± 2.79, ranged from 19 to 31) (‘Single’ group) and twenty-nine healthy participants who were in exclusive relationship (17 females, mean age 25.14 ± 9.30, from 19 to 66) (‘Relationship’ group) were screened and entered the study. One participant was excluded from the Relationship group due to issues with EEG data quality (due to extreme low frequency noise in the data that could not be effectively eliminated), which led to 28 participants (17 females, mean age 25.32 ± 9.41, from 19 to 66). All participants had normal or corrected-to-normal vision. A screening questionnaire intended to filter out individuals with neurological, psychiatric, and other medical conditions that could interfere with the experiment was used to select participants. Informed consent was obtained from all participants involved in the study. The experimental procedures were approved by the University of Essex Ethics Committee (1701) and were in accordance with the standards of the Declaration of Helsinki.

### 2.2. Preliminary Questionnaires

Participants who passed the screening procedure completed a set of online questionnaires using Qualtrics (Qualtrics, Provo, UT). These concerned the measurement of personality traits and were collected prior to the experiments. Previous research showed a potential impact on the effect of mortality salience by anxiety and depression [11,21]. We used the State-Trait Anxiety Inventory Y (STAI) and Patient Health Questionnaire 4 (PHQ-4) to measure these attributes. Importantly, we established exclusion criteria for participants with severe depression (>8 score on PHQ-4) and anxiety (outside of ±2 standard deviations (SDs) from the mean). We took this action to avoid outliers associated with the prevalence of mental health issues in UK students [41]. No participant was excluded based on these criteria.

### 2.3. EEG Recording, Pre-Processing, and Analysis

Sixty-two Ag/AgCl electrodes (Easycap, BrainProducts GmbH, Gilching, Germany) were used to record electroencephalography (EEG) (Synamps RT, Neuroscan, Compumedics). The ground was at AFz. The left earlobe was used as an active reference and the right earlobe was used as an additional recording site for off-line re-reference of scalp electrodes. The electrodes were placed according to the positions of the 10–20 International System. All the electrodes had impedance lower than 10 kΩ, and the signal was amplified and digitised at 1000 Hz.

### 2.4. Somatosensory Painful Stimulation

The BioPack^®^ STMISOLA Constant Current and Constant Voltage Isolated Linear Stimulator were used to produce the electrical stimuli. The stimulator was controlled by E-Prime^®^ 2 software and was monitored by AcqKnowledge^®^ provided by BioPack^®^. STMISOLA was used in Current mode and sent 3 square-wave pulses at 150 Hz. The amplitude of the stimuli was adjusted to the participants’ individual pain threshold (that could not overcome the stimulator’s default maximum amplitude of 85 mA). Radiation of the left-hand median nerve was stimulated. The electrodes were placed on the second metacarpal bone, the furthest possible place from the *flexor pollicis brevis* and the *lateral lumbrical* muscle of the left index finger, to minimise direct muscle stimulation over the stimulation of nociceptors in the epidermis. Participants reported a painful, sharp, needle-like sensation. To prevent participant’s distraction with the observation of their own hand being stimulated and to reduce the electrical interference with the EEG recording, the left hand of the participant was shielded from view with a cardboard baffle.

### 2.5. MindSet Manipulation

Participants were asked to answer two open-ended questions. In the mortality salience (MS) condition, they were asked to “Please briefly describe the emotions that your death arouses in you” and “Jot down, as specifically as you can, what will happen to you when you physically die and once you are dead” [11,42] for the Single group, and “Please briefly describe the emotions that your romantic partner’s death arouses in you” and “Jot down, as specifically as you can, what will happen to them when they physically die and once they are dead” for the Relationship group. In the control condition (CTRL), they were asked the same questions but framed around the failure in a very important exam [23]. There was at least a 48 h lag between the two sessions and their order was pseudo-randomised between participants (Figure 1, top).

### 2.6. Threat Manipulation

Participants were told that during the experiment each stimulus would be foreshadowed by a coloured circle (yellow or blue) (Figure 1, bottom). The colours were to signal whether the following stimulus was expected to be either a normal, “square-waved”, or a special, “sigmoid-shaped”, stimulus. Participants were told that the “sigmoid-shaped” stimulus can cause more inflammation in the skin (‘high threat’ condition) while the “square-waved” are the normal stimuli used in every other research study (‘low threat’ condition). Note that this was a cover story aimed at inducing the expectation of heightened pain for the high threat condition and potentiating the effects associated with the existential mindset manipulation. Crucially, the stimulus intensity remained the same across the entire experimental session [38]. The association between the actual colour and its meaning was pseudo-randomised between participants, and their order of assignment to participants was pseudo-randomised.

### 2.7. Anxiety State and Mood Measures

According to the classical MS design [23,43,44], we collected measurements of state anxiety and positive and negative mood at four points in the experiment: before the experiment (i.e., before mounting the EEG cap), before the mindset manipulation, after the mindset manipulation, and after the experiment. We used the state version of the State Trait Anxiety Inventory Y [45] and the Positive and Negative Affect Schedule (PANAS) [46]. As previous studies have shown, when investigating the effects of distal defences, PANAS and STAI scores should not change [5,22,23]. This is in keeping with the notion that MS-related effects only appear after the idea of death is no longer in the focus of attention [5]. Valentini et al., (2015) argued that the psychological state evoked by MS may not be consciously manifest and may better be measured by physiological changes [22,47]. According to TMT, it is the ‘potential to anxiety’ which evokes the effects of MS and not experienced anxiety [48].

### 2.8. Study Design and Procedure

We collected pain ratings and neural responses to the electrical stimuli. The dependent variables are expressed as a change from baseline (pre-mindset induction) across Mindsets (MS, CTRL) and Groups (Single and Relationship groups). We compared the effects of MS directed at the participants themselves vs. MS directed at their romantic partner on their pain perception and somatosensory brain responses.

Participants sat in front of a computer comfortably resting their left arm on the table. After the EEG cap montage, the participants completed a staircase procedure in order to identify their individual pain threshold. Participants rated their perception of the painfulness level of the stimuli during the staircase phase using a visual analogue scale (VAS). On the scale, participants rated their perception from 0 (no pain) to 100 (intolerable pain). Twenty-five, fifty, and seventy-five points were highlighted on the scale. Based on their ratings the amplitude of the stimuli changed (0–9: +1 mA; 10–19: +0.5 mA; 30–39: +0.25 mA; 40–49: +0.125 mA; 50–59: no change; 60–69: −0.125 mA; 70–89: −0.25 mA; and 90–100: −1 mA). The participants continued the assessment until they constantly rated the same current intensity between 50 and 60. With their pain threshold identified (and the stimulus amplitude increased by 0.25 mA above that threshold), participants rated 60 stimuli before and 60 stimuli after mindset manipulation. They were asked to focus on the fixation cross until they received a stimulus. They were then asked to rate their perception of the same VAS used in the staircase phase. Ten minutes of play with SUDOKU was introduced as a distraction task after mindset manipulation, as per the classical MS design [23,43,44].

### 2.9. Data Preparation and Statistical Analysis

#### 2.9.1. Sample Size and Statistical Power

Concerning the effect size, Klackl and Jonas [49] in their Methods and Materials section pointed out that “*It is unclear what effect size to expect regarding physiological activation in the mortality salience paradigm, or when comparing mortality salience with dental pain salience, because there are few precursor studies*”. As such, calculating a minimum sample size based on statistical power is less reliable for studies investigating the effects of MS. Therefore, we based our group sample size on previously published studies which involved similar measurements [50,51,52,53,54]. To further increase statistical power, we used a within-subject design and applied linear mixed-effects models [55,56].

#### 2.9.2. Data Preparation

We calculated the average state anxiety and mood scores in pre- and post-MM and the difference between them within each Mindset condition, resulting in an MS and a CTRL value for state anxiety and positive and negative mood scores.

Preprocessing of the neural data was performed in EEGLAB [57]. The EEG data were resampled to 500 Hz and re-referenced to the offline reference (right earlobe). Bandpass Butterworth FIR filter (filter order 2048) from 1 to 45 Hz was used to filter the EEG signal. Then bad electrodes were rejected (probability > 5) and independent component analysis (ICA) with the Multiple Artifact Rejection Algorithm was used to clean the signal from blinks and muscle artefacts [23]. After interpolating the previously rejected electrodes, the files were imported into Letswave 7 (www.letswave.org). We re-referenced the data to the grand average and segmented them from 1 s before the electrical stimuli to 2 s after the stimuli. We then applied the Morlet wavelet transform (Gaussian envelope at 0.15, central frequency at 3 Hz) to calculate the time-frequency representation of the epochs. We obtained an estimation of the oscillatory amplitude between 1 and 15 Hz with a frequency resolution of 0.1 Hz. We focused our analysis on the theta power (3–8 Hz) as extracted from an a priori region of interest (VROI: FC1, FCZ, FC2, C1, CZ, C2, CP1, CPZ, and CP2 electrodes [40]) and from the CZ electrode in the temporal interval from 100 to 500 ms post stimulus [23]. Previously, Valentini et al. [23] showed that the somatosensory theta response was significantly greater following MS compared to a more generic type of threat, measured at Cz. Although the stimulation paradigms were significantly different (repeated paired stimuli with constant inter-stimulus interval vs. single stimulus with variable inter-stimulus interval), we expected the independent variable and experimental design to determine similar findings. Importantly, our current analytical approach decreases the chance of a type I error by extracting the relevant brain activity from an ROI rather than just from a single electrode [58]. Each epoch was baseline corrected by calculating the event-related percentage (ER%) as ${X\prime }_{i}=\frac{\left(x_{i}-mean(baseline)\right)}{mean(baseline)}$, where ${X\prime }_{i}$ is the baseline corrected, *x_i_* is the amplitude in one frequency line at *i*^th^ time, and baseline is the average amplitude in the same frequency line −0.6–−0.2 sec pre-stimulus. We calculated the time difference of the theta power by subtracting the mean pre-MM power from each post-MM trial within participant within Mindset (Figure 2; for pre- and post-MM heatmaps and topographs, see Appendix A). We then extracted the top 10% power according to the aforementioned frequency, spatial, and temporal criteria [23]. We inspected our trials visually to avoid including any artefacts. No trials were removed.

#### 2.9.3. Data Analysis

We averaged the anxiety and mood scores collected before the experiment and pre-MM measurements (pre) separately from the scores collected post-MM and after the experiment measurements (post). Then, we calculated post–pre (Δtime) values that we analysed in our models with mindset manipulation and experimental groups as fixed factors to investigate how the participants’ levels of anxiety and positive and negative mood were affected by the experiment.
δAnxiety or Mood scores ~ Mindset * Group + (1| Participant)

Alike the theta power, we calculated a ΔVAS ratings by subtracting the mean of the pre-MM values from each post-MM rating within participant within Mindset (for instance: ΔVAS-MS_i_ = post-MS_i_ − average pre-MS). The values (mood, anxiety, VAS, and theta) were then analysed in rStudio (version 4.0.1) using linear mixed effects models from the *lme4* and *lmerTEST* packages (the latter was used to calculate *p*-values). The 95% Confidence Intervals (cIs) were calculated via the bootstrapping method (5000 repetitions). The marginal and conditional R^2^ values were calculated using the *r.squaredGLMM ()* function of the *MuMIn* package. Finally, *p*-values were calculated using Satterthwaite’s approximations. Our primary model was:ΔVAS/Theta ~ Mindset * Group * Threat + (1|Participant)

The intercept of the primary model indicates the change in time in low threat CTRL condition for the Single group. The main effect of Mindset shows how the Δ values in low threat MS condition differed compared to low threat CTRL for the Single group. The main effect of Group represents how the Relationship group differs compared with the Single group in low threat CTRL. Finally, the interaction between Mindset and Group shows how the Single group and Relationship group differ in the MS condition. We added random intercepts for each participant in both analyses.

To follow up on significant interactions, we applied secondary models on split data investigating the interactions. For example, to investigate the Mindset–Group interaction, we created an additional model only for the Relationship group with only Mindset as a fixed effect. Thus, we were able to test the effect of Mindset in the Relationship group.

## 3. Results

### 3.1. Anxiety and Mood Scores

As expected, there was no change in anxiety or negative mood scores in either Mindset condition for either group (Appendix A). Surprisingly, the positive mood scores decreased regardless of Mindset condition or groups. Our model showed that the positive mood scores decreased by 2.000 ± 0.703 (t = −2.846, 95% CI: [−3.395; −0.600], *p* = 0.005) in the CTRL condition but not in the MS condition (0.880 ± 0.790, t = 1.113, 95% CI: [−0.695; 2.402], *p* = 0.271) for the Single group. The Relationship group’s positive mood score decreased similarly to that of the Single group’s in the CTRL (−0.071 ± 0.967, t = −0.074, 95% CI: [−1.956; 1.833], *p* = 0.941) and MS conditions (0.602 ± 1.087, t = 0.554, 95%CI: [−1.579; 2.707], *p* = 0.399).

### 3.2. Pain Ratings

#### 3.2.1. Low Threat Condition

There was no change from pre- to post-MM in the Single group (β = −1.891 ± 1.274, t = −1.484, 95% CI: [−4.352; 0.578], *p* = 0.150) or in the Relationship group (β = −1.736 ± 1.120, t = −1.551, 95% CI: [−3.904; 0.500], *p* = 0.132), and the primary model confirmed that there was no difference between the groups in the CTRL condition (β = 0.154 ± 1.688, t = 0.091, 95% CI: [−3.137; 3.434], *p* = 0.927).

In the MS condition, both the Single group (β = 1.044 ± 0.469, t = 2.224, 95% CI: [0.120; 1.955], *p* = 0.026) and the Relationship group (β = 3.455 ± 0.281, t = 12.306, 95% CI: [2.902; 4.010], *p* < 0.001) experienced an increase in their pain perception compared to CTRL. Furthermore, the primary model showed that there was significantly greater pain in the Relationship group compared to the Single group (β = 2.411 ± 0.524, t = 4.598, 95% CI: [1.397; 3.433], *p* < 0.001).

#### 3.2.2. High Threat Condition

In high the threat CTRL condition, the change from pre- to post-MM VAS ratings was not significantly different in the Single group compared to the low threat CTRL condition (β = 0.369 ± 0.469, t = 0.787, 95% CI: [−0.569; 1.289], *p* = 0.432). However, this difference was significant for the Relationship group (β = 0.607 ± 0.281, t = 2.163, 95% CI: [0.039; 1.145], *p* = 0.031). Importantly, the difference between the groups was not significant (β = 0.238 ± 0.524, t = 0.454, 95% CI: [−0.773; 1.259], *p* = 0.650).

In the high threat MS condition, the significant increase found in the low threat condition was reversed for the Single group. That is, the Single group experienced a decrease in pain compared with low threat CTRL (β = −1.439 ± 0.664, t = −2.167, 95% CI: [−2.701; −0.148], *p* = 0.030). Further analysis showed that there was no difference between high threat CTRL and MS for the Singles (β = −0.395 ± 0.465, t = −0.848, 95% CI: [−1.301; 0.521], *p* = 0.396).

There was no significant difference between the high threat MS condition and the low threat MS condition for the Relationship group (β = 0.158 ± 0.397, t = 0.399, 95% CI: [−0.631; 0.965], *p* = 0.690). The subsequent follow-up model explained that, similarly to the low threat condition, there was a significant difference between high threat CTRL and high threat MS for the Relationship group, which accounted for greater pain during MS than CTRL (β = 3.613 ± 0.280, t = 12.903, 95% CI: [3.061; 4.158], *p* < 0.001).

### 3.3. Brain Activtiy

#### 3.3.1. Event-Related Theta Power at Vertex Electrode (CZ)

##### Low Threat Condition

In the low threat CTRL condition, there was a significant reduction in theta activity for the Single group (β = −1.517 ± 0.107 ER%, t = −14.140, 95% CI: [−1.726; −1.303], *p* < 0.001) as well as in the Relationship group (β = −1.186 ± 0.095 ER%, t = −12.437, 95% CI: [−1.374; −1.002], *p* = 0.012). The decrease in the Single group was significantly greater compared to the decrease in Relationship group (β = 0.332 ± 0.146 ER%, t = 2.279, 95% CI: [0.045; 0.617], *p* = 0.026).

In low threat MS, there was no significant difference in Δθ activity for the Single group (β = −0.104 ± 0.056 ER%, t = −1.868, 95% CI: [−0.212; >0.000], *p* = 0.062), but there was a significant decrease for the Relationship group (β = −0.086 ± 0.034 ER%, t = −2.514, 95% CI: [−0.152; −0.017], *p* = 0.012). The difference between the effects of Mindset in the groups was not significant (β = 0.018 ± 0.063 ER%, t = 0.288, 95% CI: [−0.107; 0.146], *p* = 0.773).

##### High Threat Condition

In the high threat CTRL condition, the decrease was significantly smaller compared to low threat CTRL condition for the Single group (β = 0.156 ± 0.056 ER%, t = 2.793, 95% CI: [0.044; 0.265], *p* = 0.005). There was no difference between high and low threat CTRL conditions in the Relationship group (β = −0.026 ± 0.034 ER%, t = −0.773, 95% CI: [−0.093; 0.041], *p* = 0.440). These differences between Threat conditions in CTRL condition were significantly different between groups (β = −0.182 ± 0.063 ER%, t = −2.879, 95% CI: [−0.311; −0.057], *p* = 0.004), as the Self group had significantly less reduction in high threat condition.

There was no difference between high and low threat MS conditions in the Single group (β = 0.002 ± 0.079 ER%, t = 0.021, 95% CI: [−0.151; 0.159], *p* = 0.983) or in the Relationship group (β = 0.047 ± 0.048 ER%, t = 0.973, 95% CI: [−0.048; 0.143], *p* = 0.331). Subsequent models showed no difference between high threat CTRL and high threat MS for the Single group (β = −0.103 ± 0.058 ER%, t = −1.785, 95% CI: [−0.217; 0.013], *p* = 0.075) or for the Relationship group (β = −0.039 ± 0.034 ER%, t = −1.139, 95% CI: [−0.104; 0.027], *p* = 0.255).

#### 3.3.2. Event-Related Theta Power at Vertex Region (VROI)

##### Low Threat Condition

In the low threat CTRL condition, there was a significant reduction in θ activity pre- to post-MM in the Single group (β = −1.294 ± 0.082 ER%, t = −15.821, 95% CI: [−1.459; −1.134], *p* < 0.001) as well as in the Relationship group (β = −1.048 ± 0.067 ER%, t = −15.579, 95% CI: [−1.175; −0.915], *p* < 0.001) (Figure 2). There was a significant difference between the two groups (β = 0.246 ± 0.105 ER%, t = 2.342, 95% CI: [0.40; 0.448], *p* = 0.023).

In low threat MS, the Δθ activity was not different from low threat CTRL in the Single group (β = −0.020 ± 0.036 ER%, t = −0.542, 95% CI: [−0.090; 0.050], *p* = 0.588), but the decrease in θ activity was significantly greater in the low threat MS condition compared to the low threat CTRL condition in the Relationship group (β = −0.080 ± 0.022 ER%, t = −3.627, 95% CI: [−0.124; −0.038], *p* < 0.001) (Figure 2). However, these differences between the Mindset conditions were not significantly different between the experimental groups (β = −0.060 ± 0.041 ER%, t = −1.471, 95% CI: [−0.143; 0.022], *p* = 0.141).

##### High Threat Condition

In high threat CTRL, there was significantly less reduction in the Single group (β = 0.137 ± 0.036 ER%, t = −3.742, 95% CI: [0.066; 0.207], *p* < 0.001), while there was no difference between Δθ activities from low and high threat CTRL conditions in the Relationship group (β = −0.014 ± 0.022 ER%, t = −0.636, 95% CI: [−0.057; 0.029], *p* = 0.524) (Figure 2). The difference between low and high threat CTRL Δθ activities was significantly smaller in the Relationship group compared to the Single group (β = −0.151 ± 0.041 ER%, t = −3.664, 95% CI: [−0.229; −0.069], *p* < 0.001).

High threat MS was not significantly different from low threat MS in either the Single (β = −0.049 ± 0.052 ER%, t = −0.946, 95% CI: [−0.148; 0.052], *p* = 0.344) or Relationship group (β = 0.037 ± 0.031 ER%, t = 1.170, 95% CI: [−0.023; 0.098], *p* = 0.242) (Figure 2).

Finally, there was no difference between the experimental groups in the high threat CTRL condition (β = 0.185 ± 0.114 ER%, t = 1.622, 95% CI: [−0.042; 0.406], *p* = 0.111). As it follows from the different effects of Threat on MS, there was no significant difference between the experimental groups in high threat MS (β = −0.065 ± 0.041, t = −1.609, 95% CI: [−0.146; 0.017], *p* = 0.108).

## 4. Discussion

Our study tested whether young adults submitted to reminders of death revealed a change in their perception and brain responses to noxious electrical stimuli, and whether these changes could be enhanced by an MS directed at their romantic partners as well as by the higher threat value of the sensory stimulation (Figure 1). According to the main tenet of the TMT [11], we expected the classical MS manipulation to induce an increase in perception and magnitude of brain responses, particularly when a more painful stimulus was expected (i.e., during the high threat condition). We also expected these effects to be greater when participants were asked to think about the death of their romantic partners.

### 4.1. Anxiety and Mood

As the TMT establishes that no explicit changes in mood and anxiety should be observed following mortality salience induction [11], we did not expect a significant difference in these ratings between mindset inductions. There was indeed no difference in state anxiety, negative mood, and positive mood between mindsets (Appendix A), thus confirming our previous findings [22,23]. However, we found a significant reduction in positive mood that was independent from the type of mindset or experimental group. This difference may simply be explained by aware reduction in positive affect due to the exposure to negative valence mindsets. Nevertheless, according to TMT, no consciously accessible changes of affect should take place whatsoever in the experimental participants (as measured by e.g., PANAS). As posited by the anxiety-buffer hypothesis, MS effects are generated from the potential for anxiety triggered by the awareness of death [48]. However, this hypothesis has been challenged and remains difficult to integrate with parsimonious models of biological responses to psychological threats [21].

It begs the question of what the underlying mechanism from which MS effects stem is. It is more likely that the effects generated by reminders of death would act through the neural paths of an overarching anxiety biological system, common to other types of symbolic and sensory threats [20,59]. This account would be able to explain mindset effects that are not specific to mortality salience. This theoretical and methodological conundrum has already been spelled out [60] and leaves researchers wondering whether both the TMT and the devices traditionally used to measure affect are not equipped to account for the hypothesised effects. However, we previously showed that self-report measures correlate with EEG changes post-MS [22], thus suggesting that while the differences in anxiety and mood between two negative mindsets may be too small to be detected with current available self-report measures, the latter could still be able to index ongoing changes associated with the experimental manipulation. If so, the ball would eventually be in the TMT field; a better explanation of MS mechanisms is required.

### 4.2. Pain Ratings

Pain intensity following MS increased in both experimental groups. However, as expected, the increase was significantly greater in the Relationship group (Figure 2, Figure 3a). This finding may be explained by the emotions of grief, loss, and even fear and anxiety induced by the idea of losing one’s own romantic partner. This finding is not surprising if we consider that the death of a loved one is an idea closely associated with brooding, depressive rumination, and distress [61]. The decrease in pain ratings during the CTRL condition can be attributed to sensory habituation. Habituation is a widely observed phenomenon in perception (and pain) experiments, especially when using single stimuli with relatively long inter-stimulus intervals, as in our study [62]. More interestingly, we found that the high threat condition was not only unable to potentiate the arousing MS effect but in fact may have led to a reduction in pain in the Self group. The manipulation of threat associated with the sensory stimuli was grounded on the notion that expectation can increase perception [63,64,65] and particularly on past empirical evidence [38] showing how the manipulation of the perceived threat value of otherwise physically identical nociceptive stimuli impacts pain perception (i.e., threatening stimuli perceived as more painful). In contrast, the decrease of pain during high threat after MS induction may rather be interpreted as the sign of threat “overload”. Previous work may suggest that if the threat value of MS is too high, the expected behavioural changes fail to materialise [11]. Crucially, although this pattern was not present in the Relationship group, we can speculate that threat manipulation is not an effective enhancer of the MS effects on pain perception.

### 4.3. Event-Related Theta Activity

Like pain ratings, Threat only affected the vertex theta activity of the Single group, thus suggesting that the interaction between the threat induced by mortality salience and the threat induced by pain anticipation may be addictive instead of multiplicative, as we assumed. Nevertheless, it is worth noting that while there was no significant effect of MS in the Single group’s neural response, the direction of the pattern is the same as the significant effects in the Relationship group: a greater decrease of theta activity post-MS, as measured from both CZ and VROI. The event-related theta synchronisation decrease following exposure to the idea of the death of their romantic partner seems to suggest that this type of induction has a greater impact on cortical arousal than exposure to the participants’ own death. Based on previous research, we would have expected a greater amplitude of the neural response to painful stimuli, as MS predicts increased bodily scanning behaviours, which would assume a less inhibited cortex [66]. Recently, Mouraux and Iannetti [67] argued that the large vertex negative and positive waves, which had been linked to the main low frequency somatosensory-related synchronisation [68], may not be a good representation of subjective pain perception after all. Our results seem to support the idea that the vertex theta response may not provide an informative account on how mortality salience affects somatosensory perception.

Importantly, the reduction of event-related theta power following MS seems in contrast with the inhibition of short-term somatosensory habituation of event-related theta synchronisation observed in past studies [18,22,23] and with the enhanced response reported with other EEG measures [18,51,69]. Notably, a few studies also reported a reduction in brain activity, as indexed by visual evoked potentials [19], reduced activity of the salience network while learning [70], and during reinforced learning [53]. In their study on reinforced learning Gao et al., [53] argued that MS may act as a dampener on slower cognitive processes, despite its effect of increased early attentional processing. According to this interpretation, the slower emotional processes linked with the distal defences may vary with methodological and individual differences.

### 4.4. Limitations and Future Directions

Our study introduces a few elements of novelty. First, we compared for the first time, within the same experimental paradigm, the effects associated with the two main MS scenarios, i.e., the subject’s own reminder of death vs. the subject’s romantic partner death. Second, we tested for the first time whether an additional manipulation aimed at altering the perceived threat of noxious somatosensory stimuli [38] would interact with the MS manipulation and heighten any observable effect. Our findings do not seem to provide a clear-cut pattern. On the one hand, we were able to confirm that the romantic partner’s MS is linked to greater effects on pain perception and neural activity, thus confirming the notion of a clear link between the fear of mortality and the need for close relationships [33,71]. On the other hand, we observed that increasing the anticipated threat value of a somatosensory stimulus did not contribute to increased MS effects on perception and brain activity.

One limitation that could account for the lack of an enhancement effect associated with the threat value of the sensory stimulus might be the lack of sufficient statistical power. The present study sample size is similar to that reported by previous studies investigating the effects of reminders of death using EEG [50,51,53,54,72,73] or magnetoencephalography [52]. However, the complexity of the research question would justify greater samples that will allow for obtaining a more robust quantification of the MS effects. This consideration, combined with the criticism conveyed by alternative theories [74,75,76] and the recent failure to replicate classical TMT findings [14,21,59], invites caution in the interpretation of our current results. Nonetheless, we recommend that future studies take up the challenge of testing means to increase the efficacy of MS manipulation, especially when attempting to quantify otherwise small changes in the brain after MS induction.

## 5. Conclusions

In conclusion, we investigated for the first time the effect of one’s romantic partner mortality salience on pain perception and EEG theta event-related activity. The manipulation was associated with, when compared to being reminded to one’s own mortality. However, this pattern was not replicated in the magnitude of the theta response. Our work highlights the impact of individual differences in classical EEG responses to threatening sensory stimuli during mortality salience manipulation while confirming the expected heightened perception associated with it.

## Figures and Tables

**Figure 1 brainsci-13-01077-f001:**
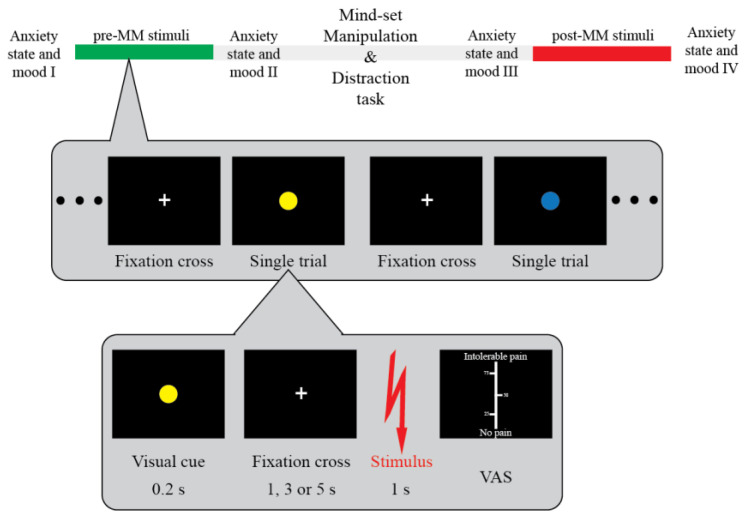
Experimental design and procedure. Participants received 60 painful electrical stimuli on the dorsum of their left hand before and 60 stimuli after being submitted to one of the questionnaires used to induce the mindset manipulation. This phase was followed by a distraction task. Each trial was foreshadowed by a coloured circle indicating the level of threat (‘high threat’, ‘low threat’; top inset) associated with the upcoming painful electrical stimuli (cf. methods for details). The electrical stimulus followed the coloured circle with a random interval (3, 4, or 5 s). Participants rated the pain associated with each electrical stimulus on a visual analogue scale (0, no pain; 100, intolerable pain). Participants were asked to fill the state anxiety and mood questionnaires at four points of the experiment: before the experiment, before the mindset manipulation (pre-MM), after the mindset manipulation, and after the experiment (post-MM).

**Figure 2 brainsci-13-01077-f002:**
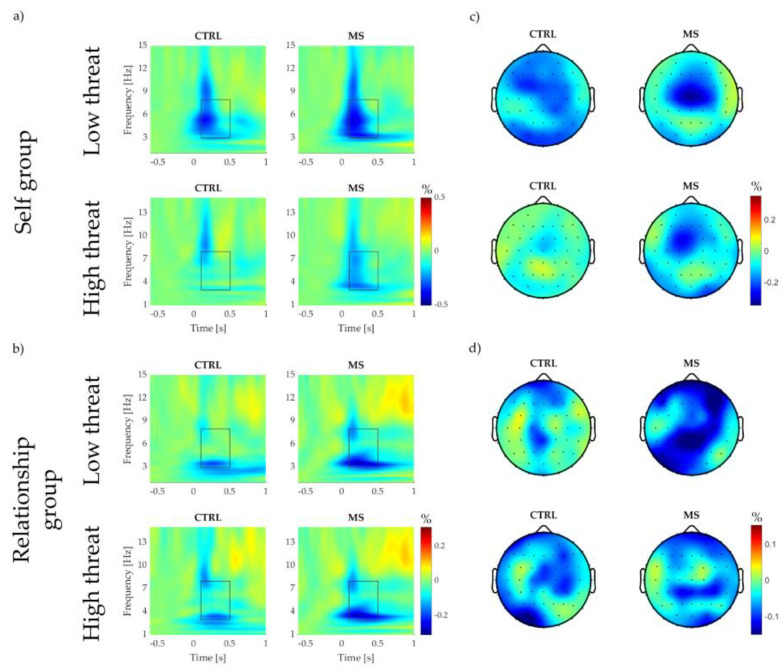
Time–frequency heatmaps (VROI) and topographies of the post- and pre-MM brain responses to somatosensory stimuli. (**a**) shows the Single group in low threat (**top** row) high threat (**bottom** row) conditions; (**b**) shows the Relationship group in low threat (**top** row) high threat (**bottom** row) conditions; (**c**) represents the topographies of the Self group from the a priori window of interest (100–500 ms post stimulus, from 3 to 8 Hz, θ activity); and (**d**) represents the topographies of the Relationship group from the a priori window of interest (100–500 ms post stimulus, from 3 to 8 Hz, θ activity).

**Figure 3 brainsci-13-01077-f003:**
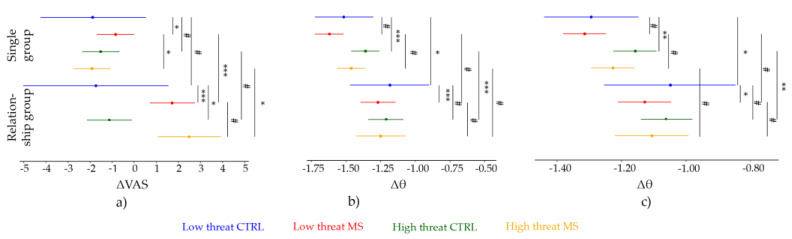
Results of the mixed-effect models. (**a**) Pain perception, (**b**) EEG theta CZ, and (**c**) EEG theta VROI. The dots represent the estimates, and the lines represent the 95% CIs. The top four lines show the data from the Single group and the bottom four represent the data from the Relationship group. Blue: Non-threatening CTRL, red: Non-threatening MS, green: Threatening CTRL, and orange: Threatening MS. The significance levels of the comparisons are represented by the vertical lines. ***—*p* < 0.001, **—*p* < 0.01, *—*p* < 0.05, #—*p* > 0.05; *p*-values were calculated using Satterthwaite’s method, and 95% CIs were calculated with the bootstrapping method using 5000 iterations.

## Data Availability

The data collected and analysed in this study can be accessed on: https://osf.io/j6qnu/.

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
