# Peer review of "Reminders of Mortality: Investigating the Effects of Different Mortality Saliences on Somatosensory Neural Activityâ€"

_brainsci, 2023, doi:10.3390/brainsci13071077_

Round 1
Reviewer 1 Report
Comments and Suggestions for Authors
The current research aimed to investigate the effects of different mortality saliences on somatosensory neural activity based on the terror management theory (TMT). They hypothesized that (1) the idea of the death of one’s romantic partner would result in larger behavioral and neural effects than those triggered by the idea of one’s own death; (2) more threatening stimuli would have been more effective in highlighting the increased physiological arousal triggered by MS effects compared with less threatening stimuli. To test these hypotheses, they presented participants with the idea of the participants’ romantic partner death as well as increased the perceived threat of somatosensory stimuli; linear mixed modelling disclosed the greater effects of MS directed at one’s romantic partner on pain perception. Moreover, the theta event-related oscillatory activity measured at the vertex of the scalp was significantly lower compared to the control condition. It is an interesting study, but I have some concerns that should be addressed before it is suitable for publication.
(1) There are too many concepts in the instruction, such as motivation, cultural views, salience network, self-esteem, anxiety, uncertainty, threat and so on. Although TMT can offer a great deal of generative hypotheses and these concepts are from these generative hypotheses, it is hard for the reader to catch up the purports of this paper. Thus, we suggest the authors to develop their hypotheses more directly.
(2) The theta event-related oscillatory is an important electrophysiological index, but it is unclear whether they select the frequency band in the instruction. It seems that the roles of theta event-related oscillatory has been confirmed in their previous studies. However, the background knowledge is not introduced in the instruction.
(3) In the 2.1 Participants, the authors said that "One participant was excluded from the Relationship group due to issues with EEG data quality", but the criterion is not unclear. Indeed, there are lots of criterion to exclude the participant in the EEG studies, such as observing the waveform, the remaining trails are too few after deleting the epochs exceeding certain values (±100μV). However, it is unclear.
(4) In EEG studies, it is appropriate to test the hypotheses by using the ROI to control the type I error. In this study, the ROIs are FC1, FCZ, FC2, C1, CZ, C2, CP1, CPZ, and CP2. Although it is also appropriate based on previous studies, it is not clear which hypotheses are tested by using the ROIs.
Reviewer 2 Report
Comments and Suggestions for Authors
Thank you for asking me to review this ms. The study is interesting and well conducted. My comments are outlined below.
How the authors ensured that the mind set manipulation took place? Anxiety (STAI) and mood (PANAS) scales were used but no significant difference was observed in the MS condition, although a decreased was observed in the control condition. The authors also discuss that no difference was to be expected after mind-set induction. This puzzled the question regarding how to ensure a successful MM.
Maybe these scales are not sensitive enough for capturing the existential anxiety?
The age range for the two groups: single (19-31 y.); exclusive relationship (19-66 y.) is not comparable, should be this a factor influencing the perception of MS? For example, young adults vs. older adults? Additionally, the age range of the individuals in the exclusive relationship is quite broad (19-66) and could it be possible that the time spent in a long-term relationship could also affect more the MS when asked to think about the death of their romantic partners? Are the authors have any indication about the length of the romantic relationship?
Participants received 60 painful stimulations before and 60 after the MM, divided in high and low threat. Was the number of trial stimulations enough to reduce the variance and enhance the signal-to-noise ratio? after cleaning and removal of artefact how was the percentage of trials included?
Both the central Cz and vertex region were considered in the analyses. However, the results from both analyses support each other, what was the rationale in choosing to conduct two separate analyses on both? Would it be enough to conduct only analyses on one, either Cz or the vertex region? Additionally, topographical maps also show activity in more frontal areas in the relationship group? Are these artifacts or are associated brain responses?
Figure 3 is a good summary picture but somehow the low quality and the small characters make difficult to read.
